# Multivariable Causal Discovery with General Nonlinear Relationships

**Patrik Reizinger**[*1]      **Yash Sharma**[1]     **Matthias Bethge**[1]

**Bernhard Schölkopf**[2]     **Ferenc Huszár**[†3]     **Wieland Brendel**[†1]

[1]University of Tübingen, Germany
[2]MPI-IS Tübingen, Germany
[3]University of Cambridge, United Kingdom

## Abstract

Today's methods for uncovering the causal relationship(s) from observational data either constrain the function class (linearity/additive noise) or the data. We make assumptions on the data to develop a framework for Causal Discovery (CD) that works for general non-linear dependencies. Similar to previous work, we use nonlinear Independent Component Analysis (ICA) to infer the underlying sources from the observed variables. Instead of using conditional independence tests to determine the causal directions, we rely on the Jacobian of the inference function; thus, generalizing LiNGAM's approach to the nonlinear case. We show that causal models resolve the permutation indeterminacy of ICA and prove that under strong identifiability, the inference function's Jacobian captures the sparsity structure of the causal graph. We demonstrate that our method can infer the causal graph on multiple synthetic data sets.

## 1 INTRODUCTION

Traditional statistical learning methods model correlations in data. Though they have achieved super-human performance in multiple fields [53, 12, 49], they have limited value in understanding cause-effect relationships. A prevalent consequence of this shortcoming is the observed tendency for models to learn shortcuts [6] (e.g., classifying objects based on their backgrounds). Conversely, *causal models* [40] construct the world according to the Independent Causal Mechanisms (ICM) principle [42], where building blocks (mechanisms) neither influence nor inform each other. Modeling temperature $T$ and altitude $A$ is a classic example [42]: changing $A$ affects $T$, but not vice versa. This independence

translates to the Directed Acyclic Graph (DAG) $A \rightarrow T$.

*Causal Discovery (CD)* describes the process of extracting causal structure from data in the form of a DAG. Having *interventional* data–such as in the form of Randomized Controlled Trials (RCTs)–is desirable as it enables answering questions of interventional nature, such as 'What will happen if variable $X$ is changed?'. However, RCTs can be costly, infeasible [4], or even unethical. Thus, developing effective CD methods reliant on *observational* data alone is of significant interest. In general, inferring the causal direction is provably impossible without additional constraints or assumptions [61]; therefore, existing methods constrain either the model class (i.e., the functions generating the observations) or the data distribution. On the model side, these constraints include linear [48, 52, 46, 62] or specific nonlinear relationships (e.g., with additive noise) [13, 44, 59, 47, 28, 38]. On the data side, assumptions include non-stationarity [35] or exchangeability [10].

CD aims to infer the ground-truth cause-effect relationships, which connects it to the *identifiability* literature, where the goal is to learn a model equivalent to the ground truth (up to indeterminacies, such as permutations or element-wise nonlinearities). An extensively studied method for learning identifiable representations is Independent Component Analysis (ICA) [2, 18], which requires that the inferred components *(sources)* are independent. Recent work has relied on NonLinear Independent Component Analysis (NLICA) [63, 15, 15, 57, 23, 20, 37, 35, 24, 8, 16, 19, 11, 29] for identifiability.

Our work builds on Monti et al. [35], which showed that NLICA can be used for CD with general nonlinear functions and observational data. Instead of using pairwise independence tests, we draw inspiration from the Linear Non-Gaussian Acyclic Model (LiNGAM) [48], which uses a weight matrix to infer the DAG of a linear causal model. We extend this approach to the nonlinear case by showing that the Jacobian of the inference function (mapping

---

*Corresponding author. Code available at: github.com/rpatrik96/nl-causal-representations

†Joint senior author.

*Accepted for the Causal Representation Learning workshop at the 38th Conference on Uncertainty in Artificial Intelligence (UAI CRL 2022).*

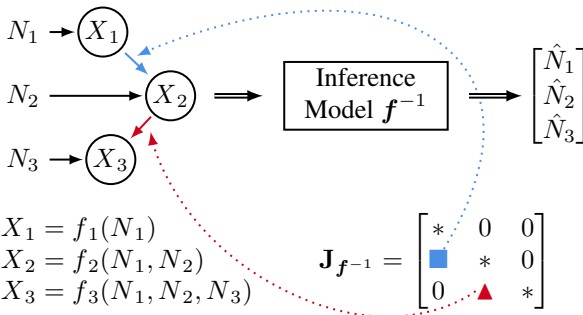

$$X_1 = f_1(N_1)$$
$$X_2 = f_2(N_1, N_2)$$
$$X_3 = f_3(N_1, N_2, N_3)$$

$$\mathbf{J}_{\boldsymbol{f}^{-1}} = \begin{bmatrix} * & 0 & 0 \\ \blacksquare & * & 0 \\ 0 & \blacktriangle & * \end{bmatrix}$$

Figure 1: **The Jacobian of the inference network $\mathbf{J}_{\boldsymbol{f}^{-1}}$ informs about the DAG**. We show that when observations $\boldsymbol{X}$ are generated from noise variables $\boldsymbol{N}$ via a general nonlinear Structural Equation Model (SEM) $\boldsymbol{f}$, then the corresponding DAG can be inferred from the Jacobian of a model that identifies $\boldsymbol{N}$ under certain assumptions on $\boldsymbol{N}$

from observations $\boldsymbol{X}$ to noise variables $\boldsymbol{N}$) captures the sparsity structure of the DAG, when strong identifiability is fulfilled [24, Def.1]. Relying on the Jacobian improves scalability, since it removes the cost of $d^2$ independence tests for a DAG with $d$ nodes. We train our model with NLICA, and show that the DAG underlying the Data Generating Process (DGP) provides an inductive bias to account for the permutation indeterminacy of NLICA.

Our **contributions** can be summarized as follows:

1. We show that causal models allow us to account for the permutation indeterminacy of ICA;
2. We prove that we can infer the DAG from the Jacobian of the inference function and also improve scalability by removing the need for independence tests.
3. We propose a multivariable CD method for general nonlinear functions from observational data;
4. We experimentally show that our proposed method can infer the DAG across multiple synthetic data sets.

## 2 BACKGROUND

Here, we describe causal models and connect their estimation to ICA. We defer technical details to Appx. A.

**Structural Equation Models (SEMs).** Given $d$-dimensional observed $\boldsymbol{X}=(X_1, \ldots, X_d)$ and noise (independent) variables $\boldsymbol{N}=(N_1, \ldots, N_d)$, their causal relationship is given by $d$ *deterministic* functional assignments [42],

$$X_i = f_i(\boldsymbol{Pa}_i, N_i) \qquad \forall i, \qquad (1)$$

where $\boldsymbol{Pa}_i \subset \boldsymbol{X}$ are the parents of $X_i$ and $f_i$ are the components of the vector-valued function $\boldsymbol{f}$. We describe the computation of $\boldsymbol{X}$ for a given $\boldsymbol{N}$ with an iterative process (denoting the iteration step with a superscript),

which is a useful concept for justifying our proposal (§ 3). Initially, $\boldsymbol{N}$ is drawn from its density. To calculate $\boldsymbol{X}$ for $\boldsymbol{N}$, the functional assignment $\boldsymbol{f}$ needs to be applied $d$ times. Namely, according to (1), each $X_i$ requires that its parents $\boldsymbol{Pa}_i$ are calculated. After sampling $\boldsymbol{N}$, only the (empty) parent sets of root nodes are calculated. Thus, the first application of $\boldsymbol{f}$ yields the $X_i$ values for such nodes. In the second iteration, the children of root nodes can be calculated (since we have all parents from the first iteration), and so on. This yields an iterative algorithmic formulation of the SEM, describing the computational graph given by the DAG as:

$$\boldsymbol{X} = \boldsymbol{X}^d = \boldsymbol{f}^{(d)}\left(\boldsymbol{X}^0, \boldsymbol{N}\right), \qquad (2)$$

where $\boldsymbol{X}^0$ is the initial value (w.l.o.g., we assume $\boldsymbol{X}^0 = \boldsymbol{0}$, since calculating the functional assignments will overwrite every $X_i$). As in most previous works [55, Table 1], we assume *no confounders* (all variables are observed) and *faithfulness* (loosely speaking, the coefficients/functions will not cancel an edge, cf. Assum. A.1).

**Causal Discovery (CD).** In CD, the data is assumed to be generated by a causal process, and the aim is to infer the corresponding DAG, which enables reasoning about interventions (without the DAG, the joint distribution $p(\boldsymbol{N})$ only admits observational queries) [42, 41]. Algorithmic approaches include combinatoric search [48, 13, 14, 21, 34, 43, 50, 55], continuous optimization [62, 30, 56, 39, 55], and neural networks [59, 38, 25, 58, 7, 22, 55, 27, 36]—we focus on the latter. Zhang et al. [61] proved that identifying the causal direction in a general SEM is impossible without constraints on the function class and/or data distribution. Functional constraints can include linear [48, 62], additive nonlinear ($X_i = f_i(\boldsymbol{Pa}_i) + N_i$) [13, 38, 28, 44], or affine nonlinear ($X_i = f_i(\boldsymbol{Pa}_i) + h_i(N_i)$) [25, 47] models. Regarding the data distribution, some models require access to interventions [1, 45, 31]; others assume that $\boldsymbol{N}$ is Gaussian [22, 28] or non-Gaussian [48]; or require non-stationarity [35], exchangeability [10], or discreteness [45] of $\boldsymbol{N}$. Our work was inspired by [35], which provides a bivariate CD method for general nonlinear functions and non-stationary data. The authors leverage recent results in NLICA (cf. next section for details) to identify the causal direction. Although they demonstrate applicability to multivariable problems, the use of pairwise independence tests constrains scalability. In this work, we extend these results with a more scalable, end-to-end solution. For this purpose, we draw inspiration from LiNGAM [48]. Assuming that the inference model learns to map observations to latents (i.e., it "inverts" the SEM), we illustrate how the weight matrix is used to extract the DAG for a linear SEM in the following example.

**Example 1** (Motivating example for linear SEMs). *Assume a linear causal model with three variables, the DAG $X_1 \rightarrow X_2 \rightarrow X_3$, and functional relationships: $X_1 = N_1$; $X_2 = aX_1 + N_2$; $X_3 = bX_2 + N_3 : a, b \in \mathbb{R} \setminus \{0\}$. The DGP generates samples according to the DAG and has the*

*matrix form on the left—we focus on the elements below the main diagonal as for recovering the DAG, only the paths (i.e., series of directed edges) between $X_i$ and $X_j$ are required and the main diagonal expresses the $N_i - X_i$ edges. Inverting the DGP with an inference model (i.e., expressing $N_i$ as a function of $X_j$) yields the matrix on the right with elements below the main diagonal capturing the DAG's $X_i - X_j$ edges (as shown by color coding):*

$$\begin{bmatrix} X_1 \\ X_2 \\ X_3 \end{bmatrix} = \begin{bmatrix} 1 & 0 & 0 \\ a & 1 & 0 \\ ab & b & 1 \end{bmatrix} \begin{bmatrix} N_1 \\ N_2 \\ N_3 \end{bmatrix} ; \begin{bmatrix} N_1 \\ N_2 \\ N_3 \end{bmatrix} = \begin{bmatrix} 1 & 0 & 0 \\ -a & 1 & 0 \\ 0 & -b & 1 \end{bmatrix} \begin{bmatrix} X_1 \\ X_2 \\ X_3 \end{bmatrix}$$

This is the motivation for LiNGAM to infer the DAG from a weight matrix [48]—we use the same insight in the nonlinear case (cf. § 3) on the Jacobian of the inference model. As the Jacobian is a local property, we will reason about the *absolute value of the maximum Jacobian,* where the maximum is taken over the input space.

**DAG equivalence.** To justify using the Jacobian of the inference network $\boldsymbol{f}^{-1}$, akin to LiNGAM's use of a weight matrix, we first connect the DAG and $\mathbf{J}_{\boldsymbol{f}^{-1}}$ via fundamental concepts from graph theory. The *adjacency matrix* $\boldsymbol{\mathcal{A}}$ of a graph with $d$ nodes is a binary $d \times d$ matrix where each matrix element indicates the presence, or absence, of an edge between a pair of nodes $X_i, X_j$ (Defn. A.4). The *connectivity matrix* of a graph with $d$ nodes is a binary $d \times d$ matrix where each matrix element indicates the presence, or absence, of a *path* (i.e., series of directed edges) between two nodes $X_i, X_j$ (Defn. A.5). For DAGs, both $\boldsymbol{\mathcal{A}}$ and $\boldsymbol{\mathcal{C}}$ are *strictly lower-triangular*—this is why we considered only the elements below the main diagonal in Ex. 1.

The inference network $\widehat{\boldsymbol{f}}^{-1}$ generally differs from the true inverse of $\boldsymbol{f}$ up to indeterminacies (e.g., scaling, permutation, sign flips, element-wise transformations) [18, 23, 63]. Furthermore, the main diagonal of $\mathbf{J}_{\boldsymbol{f}^{-1}}$ has non-zero elements (Ex. 1). Thus, we describe the relationship between $\mathbf{J}_{\widehat{\boldsymbol{f}}^{-1}}$ and $(\mathbf{I}_d - \boldsymbol{\mathcal{A}})$ for a DAG via *structural equivalence,* and investigate its symmetries (∘ denotes composition):

**Definition 1** ($\sim_{DAG}$)**.** *Two matrices $\mathbf{S}, \mathbf{R}$ are structurally equivalent if $(\mathbf{S})_{ij} = 0 \iff (\mathbf{R})_{ij} = 0 : \forall i, j$; with the properties:*

  (i) *$\mathbf{D}$-invariance: a non-singular diagonal matrix $\mathbf{D}$ preserves the sparsity structure; thus, $(\mathbf{D} \circ \mathbf{S}) \sim_{DAG} \mathbf{S}$*
  (ii) *$h_0$-invariance: for zero-preserving transformations $h_0: (h_0(\mathbf{S}))_{ij} = 0 \iff (\mathbf{S})_{ij} = 0$ then $h(\mathbf{S}) \sim_{DAG} \mathbf{S}$*
  (iii) *$\pi$-equivariance: a permutation $\pi$ affects the positions of zeros; thus, both operands need to be permuted with the same $\pi$ to maintain $\sim_{DAG}$, i.e., $\mathbf{S} \sim_{DAG} \mathbf{R} \iff (\pi \circ \mathbf{S}) \sim_{DAG} (\pi \circ \mathbf{R})$,*
  (iv) ***Transitivity:** $\mathbf{S} \sim_{DAG} \mathbf{P} \wedge \mathbf{P} \sim_{DAG} \mathbf{R} \implies \mathbf{S} \sim_{DAG} \mathbf{R}$*
  (v) ***Commutativity:** $\mathbf{S} \sim_{DAG} \mathbf{R} \iff \mathbf{R} \sim_{DAG} \mathbf{S}$.*

$\mathbf{S} \sim_{DAG} \mathbf{R}$ thus implies the matrices have the same sparsity structure. Thereby, if $\mathbf{S}$ and $\mathbf{R}$ are adjacency matrices, they describe the same DAG.

**Identifiability and ICA.** Independent Component Analysis (ICA) [2, 18] models the observed variables $\boldsymbol{X}$ as a mixture of *independent* variables $\boldsymbol{N}$ via a deterministic function $\boldsymbol{f}$, and focuses on defining models that are *identifiable*—i.e., $\boldsymbol{N}$ can be recovered up to indeterminacies (e.g., scaling, permutation, sign flips, element-wise transformations). Since this is provably impossible in the nonlinear case without further assumptions [3, 17, 32], recent work has focused on incorporating *auxiliary* variables [20, 8, 23, 5], exploiting temporal structure in the data [16, 15, 11, 37, 35, 19, 26, 63], or restricting the model class [48, 13, 60, 9] . Several works have related (nonlinear) ICA to SEM estimation [9, 35, 48, 54] by inverting the DGP—i.e., estimating $\boldsymbol{f}^{-1}$ with an *inference model.*

## 3 PROPOSED METHODS

We propose an extension of LiNGAM [48] to general nonlinear relationships. We require strong identifiability [24, Def.1] of the inference function $\boldsymbol{f}^{-1}$ for extracting the DAG via the Jacobian $\mathbf{J}_{\boldsymbol{f}^{-1}}$ from observational data. First, we observe that by assuming a DAG for the DGP, the permutation indeterminacy of ICA can be accounted for (cf. Appx. A.1 for the origin of the two permutations) —we then exploit this in Prop. 1 to prove that strongly identified models fulfil $\mathbf{J}_{\boldsymbol{f}^{-1}} \sim_{DAG} (\mathbf{I}_d - \boldsymbol{\mathcal{A}})$.

**Lemma 1** (DAG DGPs with unique $\pi$ provide additional information for resolving the permutation ambiguity of ICA)**.** *When $\boldsymbol{f}$ describes a DAG, then the permutation indeterminacy of ICA $\pi_{\mathrm{ICA}}$ can be resolved uniquely, even with unknown but unique causal ordering $\pi$.*

*Proof.* The unknown causal ordering $\pi$ of $N_i$ implies the right-multiplication of $\mathbf{J}_{\boldsymbol{f}^{-1}}$ with $\pi^{-1}$, whereas the permutation indeterminacy of ICA implies the left-multiplication with $\pi_{\mathrm{ICA}}$, yielding the following estimated Jacobian:

$$\mathbf{J}_{\widehat{\boldsymbol{f}}^{-1}} = \pi_{\mathrm{ICA}} \circ \mathbf{J}_{\boldsymbol{f}^{-1}} \circ \pi^{-1}, \tag{3}$$

where $\pi_{\mathrm{ICA}}$ and $\pi^{-1}$ are not necessarily the same. As SEMs have a lower-triangular Jacobian and we assume that $\pi$ is unique, this inductive bias on $\mathbf{J}_{\boldsymbol{f}^{-1}}$ provides an unsupervised means to resolve $\pi_{\mathrm{ICA}}$ and $\pi^{-1}$ and recover $\mathbf{J}_{\boldsymbol{f}^{-1}}$ from the estimated $\mathbf{J}_{\widehat{\boldsymbol{f}}^{-1}}$. $\square$

Relying on Lemma 1 and the properties of $\sim_{DAG}$, we prove that $\mathbf{J}_{\boldsymbol{f}^{-1}}$ can be used to extract the DAG for general nonlinear functions (akin to the linear case shown in Ex. 1):

**Proposition 1** ($\mathbf{J}_{\widehat{\boldsymbol{f}}^{-1}} \sim_{DAG} (\mathbf{I}_d - \boldsymbol{\mathcal{A}})$)**.** *The inference network Jacobian $\mathbf{J}_{\widehat{\boldsymbol{f}}^{-1}}$ is structurally equivalent to $(\mathbf{I}_d - \boldsymbol{\mathcal{A}})$*

*if $\boldsymbol{f}^{-1}$ is strongly identified [24, Def.1] up to scalings, sign flips, permutations, and zero-preserving transformations.*

*Proof.* The proof consists of two steps: 1) leveraging the iterative formulation of the SEM (2), proving that $\mathbf{J}_{\boldsymbol{f}^{-1}} \sim_{DAG} (\mathbf{I}_d - \boldsymbol{\mathcal{A}})$ and 2) relying on the properties of $\sim_{DAG}$ and Lemma 1, showing $\mathbf{J}_{\boldsymbol{f}^{-1}} \sim_{DAG} \mathbf{J}_{\widehat{\boldsymbol{f}}^{-1}}$.

We start by formulating $\mathbf{J}_{\boldsymbol{f}}$ (recall that $\boldsymbol{X} = \boldsymbol{X}^d$) based on the iterative SEM expression (2):

$$\mathbf{J}_{\boldsymbol{f}} = \tfrac{\partial \boldsymbol{X}^d}{\partial \boldsymbol{N}} = \mathbf{A} \tfrac{\partial \boldsymbol{X}^{d-1}}{\partial \boldsymbol{N}} + \mathbf{B} \qquad (4)$$

$$\mathbf{A} := \tfrac{\partial \boldsymbol{f}(\boldsymbol{X}^{d-1}, \boldsymbol{N})}{\partial \boldsymbol{X}^{d-1}}; \ \mathbf{B} := \tfrac{\partial \boldsymbol{f}(\boldsymbol{X}^{d-1}, \boldsymbol{N})}{\partial \boldsymbol{N}}, \qquad (5)$$

where $\mathbf{A}$ describes the $X_i - X_j$ edges in the DAG (i.e., $\mathbf{A} \sim_{DAG} \boldsymbol{\mathcal{A}}$), $\mathbf{B}$ is diagonal (as the $\boldsymbol{X}^{d-1}$ values are fixed) and both $\mathbf{A}, \mathbf{B}$ are independent from $t$ (superscript).

Realizing that (4) gives us a recursive formula, and recalling that $\boldsymbol{X}^0 = \mathbf{0}$, we can unroll (4) iteratively for $t = d - 1, d - 2, \ldots, 0$:

$$\mathbf{J}_{\boldsymbol{f}} = \mathbf{A} \tfrac{\partial \boldsymbol{X}^{d-1}}{\partial \boldsymbol{N}} + \mathbf{B} = \mathbf{A} \left[ \mathbf{A} \tfrac{\partial \boldsymbol{X}^{d-2}}{\partial \boldsymbol{N}} + \mathbf{B} \right] + \mathbf{B} \qquad (6)$$

$$= \mathbf{A} \left[ \mathbf{A} \left[ \ldots \left[ \mathbf{A} \underbrace{\tfrac{\partial \boldsymbol{X}^0}{\partial \boldsymbol{N}}}_{=\mathbf{0}} + \mathbf{B} \right] \right] + \mathbf{B} \right] + \mathbf{B} \qquad (7)$$

$$= \sum_{i=0}^{d-1} \mathbf{A}^i \mathbf{B} = (\mathbf{I}_d - \mathbf{A})^{-1} \mathbf{B}, \qquad (8)$$

where the last equality expresses the sum of the geometric series with elements $\mathbf{A}^i$ (the sum is finite as $\mathbf{A}$ is strictly lower triangular). By invoking the inverse function theorem, we can express $\mathbf{J}_{\boldsymbol{f}^{-1}}$:

$$\mathbf{J}_{\boldsymbol{f}^{-1}} = \mathbf{J}_{\boldsymbol{f}}^{-1} = \mathbf{B}^{-1} (\mathbf{I}_d - \mathbf{A}). \qquad (9)$$

$\mathbf{J}_{\boldsymbol{f}^{-1}} \sim_{DAG} (\mathbf{I}_d - \boldsymbol{\mathcal{A}})$ follows as $\mathbf{A} \sim_{DAG} \boldsymbol{\mathcal{A}}$ and $\mathbf{B}$ is diagonal (the invariance of $\sim_{DAG}$ follows from Prop. 1(i)). For proving that $\mathbf{J}_{\widehat{\boldsymbol{f}}^{-1}} \sim_{DAG} (\mathbf{I}_d - \boldsymbol{\mathcal{A}})$, we need $\mathbf{J}_{\boldsymbol{f}^{-1}} \sim_{DAG} \mathbf{J}_{\widehat{\boldsymbol{f}}^{-1}}$ (Prop. 1(iv)), which requires us to account for all indeterminacies of strong identifiability: i) Prop. 1(i) accounts for scalings and sign flips; ii) Prop. 1(ii) for zero-preserving transformations; and iii) Prop. 1(iii) for permutations, which can be extracted as shown in Lemma 1. $\qquad \square$

Prop. 1 implies that we can extract the DAG when $\boldsymbol{f}^{-1}$ can be strongly identified [24, Def.1]—i.e., we can reason about interventions (cf. § 2). We note that if $\mathbf{B} = \mathbf{I}_d$, then (9) describes Additive Noise Models (ANMs) [13], whereas when additionally $\mathbf{A}$ is constant, we recover LiNGAM [48].

**Description of the algorithm for CD and determining $\pi$.** We propose a two-step approach for extracting the DAG from observational data (Alg. 1) for general nonlinear $\boldsymbol{f}$:

---

**Algorithm 1** Algorithm for multivariable CD and determining the causal order $\pi$

---

**Input:** dataset $D$, network parameters $\theta$, Sinkhorn networks $\mathbf{S}_{\mathrm{ICA}}, \mathbf{S}_\pi$
Initialize $\theta$
**while** $\mathcal{L}_{CL}$ not converged **do**
    sample batch from $D$
    calculate $\mathcal{L}_{CL}$
    update $\theta$
**end while**
extract $\mathbf{J}_{\widehat{\boldsymbol{f}}^{-1}}$
**while** $\mathcal{L}_\pi$ not converged **do**
    $\mathbf{K} = \left| \mathbf{S}_{\mathrm{ICA}} \mathbf{J}_{\widehat{\boldsymbol{f}}^{-1}} \mathbf{S} \right|$
    $\mathcal{L}_\pi = \sum_{i,j} \left[ \alpha_d (\mathbf{K})_{ii}^{-1} + \alpha_u (\mathbf{K})_{i<j} - \alpha_l (\mathbf{K})_{i \geq j} \right]$
    update $\mathbf{S}_{\mathrm{ICA}}, \mathbf{S}_\pi$
**end while**

---

1. we estimate $\boldsymbol{f}^{-1}$ with an inference model that ensures (strong) identifiability,
2. we account for the ordering to resolve the permutation indeterminacy.

Regarding the second step, the training objective for learning the permutations in (3) is inspired by LiNGAM [48] and leverages the observation that in SEMs, the ground-truth Jacobian $\mathbf{J}_{\boldsymbol{f}^{-1}}$ is lower-triangular:

$$\mathcal{L}_\pi = \sum_{i,j} \left[ \alpha_d (\mathbf{K})_{ii}^{-1} + \alpha_u (\mathbf{K})_{i<j} - \alpha_l (\mathbf{K})_{i \geq j} \right] \quad (10)$$

$$\mathbf{K} := \left| \mathbf{S}_{\mathrm{ICA}} \mathbf{J}_{\widehat{\boldsymbol{f}}^{-1}} \mathbf{S}_\pi \right|, \qquad (11)$$

where $\mathbf{S}_{\mathrm{ICA}}, \mathbf{S}_\pi$ are doubly-stochastic matrices, $(\mathbf{K})_{i \geq j}$ are the lower-, $(\mathbf{K})_{i < j}$ *strictly* upper-triangular elements of $\mathbf{K}$, and $\alpha_{\{d,l,u\}} > 0$. $\mathcal{L}_\pi$ encourages $\mathbf{K}$ to be lower-triangular by simultaneously: maximizing i) the sum of the main diagonal; ii) the lower-triangular part; while also iii) minimizing the stricly-upper triangular part of $\mathbf{K}$.

## 4 EXPERIMENTS

**Experimental setup.** To (strongly) identify the SEM (quantified by Mean Correlation Coefficient (MCC) [15]), we use contrastive NLICA [63] to estimate $\widehat{\boldsymbol{f}}^{-1}$, and satisfy the assumptions on the DGP underlying the proof of identifiability [63, Thm. 6]) accordingly: the latent space is a hyperrectangle in $\mathbb{R}^d$, the marginal $p(\boldsymbol{N})$ is uniform, the conditional $p(\tilde{\boldsymbol{N}}|\boldsymbol{N})$ is Laplace, $\boldsymbol{X}$ is generated by a smooth and bijective mapping; and the contrastive loss uses the same metric as the conditional, which is $L_1$ for our case (Assum. B.1). Our architecture for the inference model is the same MultiLayer Perceptron (MLP), as in [63] (Tab. 3). To account for the permutation indeterminacies, we use two Sinkhorn networks [33], which are differentiable models for learning doubly-stochastic matrices. We observed that set-

ting the lowest $d(d-1)/2$ elements to zero and converting the resulting $\mathbf{K}$ matrix to binary often helped the convergence of the Sinkhorn networks. Moreover, instead using max to aggregate the different Jacobians over the batch, we found using the mean operator more stable in practice.

We experiment with three DGPs: i) linear and ii) nonlinear SEMs (in the simple form of $\mathbf{X} = \boldsymbol{f}(\mathbf{WN})$, as well as iii) MLPs with triangular weight matrices (as used in [35]). In all cases, the nonlinear activations are leaky ReLUs (with a slope of $0.25$ for the SEMs and $0.1$ for the triangular MLPs). Additionally, we ensure that the ordering of $N_i$ is unique (all cases), and that the DGP weights are $\gg 0$ (for the SEM DGPs) as otherwise we would be unable to distinguish weak connections from small elements in the Jacobian. That is, the estimate of a weak connection could be the same order of magnitude as the estimate of a zero element due to finite numerical precision—we do not enforce this property for the triangular MLPs to compare to the results of [35], where such modification was not present. For the SEM DGPs, we sample 6 different orderings and 5 seeds for each problem dimensionality $\{3; 5; 8\}$. For the triangular MLP, we use $d = 6$ to compare to [35, Fig. 2] and vary the number of layers in the mixing. We measure learning the correct ordering by the ordering accuracy ($\mathrm{Acc}_\pi$)—i.e., ratio when $\mathbf{S}_\pi$ inverts $\pi$. We also report the accuracy (Acc) and the Structural Hamming Distance (SHD) (we use $1\mathrm{e}{-3}$ as the threshold in all scenarios) for inferring the edges of the DAG, as is standard practice in the literature[28, 35, 45, 55]. We use the linear and nonlinear SEM DGPs to showcase that our method can infer the DAG while also learning the correct ordering. Then, we compare to the methods reported in [35], which unlike our proposal, assume that $\pi$ is the identity.

**Results.** Tab. 1 demonstrates that our method works almost perfectly in the linear case, whereas its performance is slightly worse in the nonlinear case in terms of accuracy, SHD and MCC. This means that most edges are inferred correctly and identifiability is achieved. Nonetheless, accounting for both the ICA permutation indeterminacy and $\pi$ degrades with increasing $d$. Nonetheless, erroneous solutions resulting from optimization issues (the most frequent problem according to our observations) can be simply filtered out: in this case the doubly stochastic matrices usually do not converge to a permutation matrix. Inspecting their elements or automatically rejecting such solutions based on their entropy is straightforward (permutation matrices have minimal entropy among doubly stochastic matrices, so higher entropy means to a suboptimal solution).

Tab. 2 summarizes our results with the triangular MLP of [35]. Despite having small weights in the ground truth Jacobian $\mathbf{J}_{\boldsymbol{f}^{-1}}$, our method was able to infer most edges in the DAG. Importantly, the resulting accuracies are larger than for NonSENS [35]. Moreover, our method has the advantage of simultaneously inferring all edges based on the structure of $\mathbf{J}_{\widehat{\boldsymbol{f}}^{-1}}$—thus, it does not require $d^2$ pairwise independence

Table 1: Results for linear and nonlinear SEMs. Mean Correlation Coefficient (MCC) measures identifiability, Acc stands for accuracy (the subscript $\pi$ denotes the accuracy of accounting for the causal ordering $\pi$), and SHD is the Structural Hamming Distance

| DGP | $d$ | MCC | $\mathrm{Acc}_\pi$ | Acc | SHD |
|---|---|---|---|---|---|
| | 3 | 1. | 1. | 1. | 0. |
| Lin. SEM | 5 | 1. | 0.966 | 1. | 0.0013 |
| | 8 | 1. | 1. | 1. | 0. |
| | 3 | 1. | 1. | 1. | 0. |
| Nl. SEM | 5 | $0.971 \pm 0.07$ | 0.828 | 0.974 | 0.0262 |
| | 8 | $0.987 \pm 0.03$ | 0.793 | 0.968 | 0.0318 |

test for a DAG with $d$ nodes.

Table 2: Results for the triangular MLP from [35] with $d = 6$. # Layers denotes the number of layers in the mixing

| # Layers | MCC | Acc | SHD |
|---|---|---|---|
| 1 | 1. | 1. | 0. |
| 2 | 0.999 | 1. | 0.0056 |
| 3 | $0.932 \pm 0.09$ | 0.9 | 0.1 |
| 4 | $0.833 \pm 0.01$ | 0.817 | 0.1833 |
| 5 | $0.848 \pm 0.02$ | 0.839 | 0.1611 |

## 5 DISCUSSION

We introduced a two-step process to leverage strong identifiability for inferring the DAG of multivariable causal models with general nonlinear functions. Our method uses the Jacobian of the inference function (mapping from observables to independent variables) and can be thought as a generalization of LiNGAM to the nonlinear case. We prove that this Jacobian captures the sparsity structure of the DAG, and show that by working with causal models, we can resolve the permutation indeterminacy of ICA under certain assumptions. Since we do not use conditional independence tests, but learn the causal ordering with Sinkhorn networks, our method provides an end-to-end solution for CD and avoids the cost of exponentially many independence tests. We experimentally demonstrate that our proposal can infer the DAG in multiple synthetic data sets.

**Limitations.** Our theory requires the guarantees of strong identifiability but not the use of a specific (NLICA) algorithm. Though our experiments demonstrate that fulfilling strong identifiability is sufficient for CD, we do not vary the NLICA algorithm. Our method's applicability is limited for inferring weak edges, similar to [48, 52, 46, 28].

## Author Contributions

Wieland Brendel initiated the project. Ferenc Huszár and Patrik Reizinger derived the theory, Patrik Reizinger and Yash Sharma ran the experiments and all authors wrote the paper.

## Acknowledgements

The authors would like to thank Ricardo Pio Monti and Scott W. Linderman for helpful correspondence. Wieland Brendel acknowledges financial support via an Emmy Noether Grant funded by the German Research Foundation (DFG) under grant no. BR 6382/1-1. Wieland Brendel is a member of the Machine Learning Cluster of Excellence, EXC number 2064/1 – Project number 390727645. The authors thank the International Max Planck Research School for Intelligent Systems (IMPRS-IS) for supporting Yash Sharma and Patrik Reizinger. Patrik Reizinger acknowledges his membership in the European Laboratory for Learning and Intelligent Systems (ELLIS) PhD program

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

## A SEMS

**Definition A.1** (SEM). *A SEM describes causal relationships via a set of structural assignments [42]:*

$$X_i := f_i\left(\boldsymbol{Pa}_i, N_i\right), \qquad \forall i \in \mathcal{I} = \{,\dots,d\}, \quad (12)$$

*where $X_i$ are the endogenous, $N_i$ the exogenous/noise variables, $\boldsymbol{Pa}_i \subseteq \boldsymbol{X} \setminus \{X_i\}$ denotes the parent set of $X_i$, $\mathcal{I}$ the set of indices, and $f_i$ the mappings.*

**Definition A.2** (Reduced form of SEM). *The reduced form of the SEM expresses all $X_i$ as a function of only the $N_i$ variables, i.e.:*

$$X_i := f_i\left(\boldsymbol{N}^i\right), \qquad \forall i \in \mathcal{I} = \{0,\dots,d-1\}, \quad (13)$$

with the same notation as in Defn. A.1, slightly abusing $f_i$ and denoting a subset of $\boldsymbol{N}$ by $\boldsymbol{N}^i \subseteq \boldsymbol{N}$.

**Definition A.3** (Causal ordering). *The causal ordering $\pi$ is a bijective automorphism on the index set $\mathcal{I}$. Namely, $\pi : \mathcal{I} \to \mathcal{I}$ so that $\forall X_i \neq X_j$, it holds that if $\pi(i) < \pi(j) \implies X_j \notin \boldsymbol{Pa}_i$.*

The definition means that only a node with a smaller index in $\pi$ can be a parent of a node with a larger index. Note that though $X_i$ *can* be a parent of $X_j$, it is not necessary, but $X_j$ *cannot* be a parent of $X_i$. Multiple orderings may exist, e.g. if there are multiple $X_i$ so that they only have a single parent. $\pi$ helps to have a unique description of the edges in the graph. Namely, if the edges are organized in the adjacency matrix $\mathcal{A}$ accoridng to $\pi$, then $\mathcal{A}$ will be strictly lower triangular.

**Definition A.4** (Adjacency matrix). *The adjacency matrix $\mathcal{A}$ is a binary $d \times d$ matrix, where $\mathcal{A}_{ij} = 1 \iff X_j \in \boldsymbol{Pa}_i$. The rows of $\mathcal{A}$ are ordered by $\pi$; thus, $\mathcal{A}$ is strictly lower-triangular.*

$\mathcal{A}$ only describes the edges of the DAG, which gives the direct cause-effect relationships. Nodes can be influence each other via paths (i.e., a set of directed edges that can be traversed between the two nodes), which can be described by the connectivity matrix $\mathcal{C}$

**Definition A.5** (Connectivity matrix). *The connectivity matrix $\mathcal{C}$ is a binary $d \times d$ matrix, where $\mathcal{C} = 1 \iff \exists p : X_j \to \cdots \to X_i$. $\mathcal{C} = \sum_{k=1}^{d} \mathcal{A}^k$. The rows of $\mathcal{C}$ are ordered by $\pi$; thus, $\mathcal{C}$ is strictly lower-triangular.*

**Assumption A.1** (SEM assumptions). *We assume that the causal DGP fulfils:*

- (i) *(1) describes a DAG*
- (ii) *$N_i$ are jointly independent*
- (iii) *There are no hidden confounders (faithfulness/stability), i.e., all*
- (iv) *$\pi$ is unique*
- (v) *Each $f_i$ is a homomorphism (but they can be general nonlinear functions)*

Requiring a unique $\pi$ is a simplifying assumptions that to avoid ambiguities when presenting results, so it is *without loss of generality*

**Definition A.6** (DGP with known $\pi$). *The DGP is described by the SEM, when $\pi$ is known. I.e., the flow of information is: $\boldsymbol{N} \xrightarrow{SEM} \boldsymbol{X}$.*

**Definition A.7** (DGP with unknown $\pi$). *The DGP with unknown $\pi$ is given by the SEM, and by a permutation matrix $\pi$ (with a slight abuse of notation) applied to $\boldsymbol{X}$. I.e., the flow of information is: $\boldsymbol{N} \xrightarrow{SEM} \boldsymbol{X} \xrightarrow{\pi} \hat{\boldsymbol{X}}$.*

**Lemma A.1** ( $\mathbf{J_f} \sim_{DAG} (\mathbf{I}_d + \mathcal{C})$ ). *Given Assum. A.1, the partial derivatives of $f_i$ w.r.t. $N_j$ provide information about $\mathcal{C}$, as*

$$(\mathbf{J_f})_{kl} = \max_{N_k} \left| \frac{\partial f_l}{\partial N_k} \right| = 0 \iff \nexists X_k \to \cdots \to X_l$$

*We emphasize that the derivatives are also non-zero in the case of indirect paths, i.e., when $\exists X_i \in p : i \neq k, l$. Furthermore, the strictly lower triangular part of $\mathbf{J_f}$ has the describes the same DAG as $\mathcal{C}$–or equivalently, $\mathbf{J_f} \sim_{DAG} (\mathbf{I}_d + \mathcal{C})$.*

### A.1 WHY ARE THERE TWO PERMUTATION INDETERMINACIES IN Lemma 1?

In this section, we elaborate on the need to account for *two permutations* in Lemma 1: besides the well-understood indeterminacy coming from NLICA [17], the unknown causal order of $N_i$ also implies a permutation (Defn. A.7). Namely, a SEM with unknown causal ordering can be described as i) applying the SEM equations, ii) followed by a permutation matrix $\pi$. This implies a right-multiplication of $\mathbf{J_{f^{-1}}}$ with $\pi^{-1}$ to extract the original causal ordering.

Accounting for the causal ordering is, to the best of our knowledge, only found in [48]. Binary CD methods such as [35] alleviate this step as they work on an edge-by-edge basis. Other non-ICA-base methods can also avoid this step since the DAG is *invariant* to changes in the causal ordering –meaning that reordering $X_i$ in the observation vector $\boldsymbol{X}$ (cf. Defn. A.7) does not affect the edges of the graph. However, to resolve the permutation indeterminacy of ICA, we need to account for the causal ordering, since only then can the Jacobian be lower-triangular. Although extracting a lower-triangular Jacobian is easier to interpret and potentially better suited, e.g., as a building block of causal representation learning (since the causal ordering of $N_i$ is always the same), our method extracts the DAG even without resolving these indeterminacies.

## B EXPERIMENTAL DETAILS

**Assumption B.1** (NLICA assumptions). *We assume the setting of [63], specifically that of Thm. 6, under which, an encoder which minimizes a contrastive loss was proven to estimate the noise variables (often referred to as "sources" in the ICA literature) up to a composition of input independent permutations, sign flips, and rescaling. For completeness, we restate the assumptions below:*

- (i) *the space of sources/latent/noise variables, is a convex body in $\mathbb{R}^d$, i.e. a hyperrectangle/cube.*
- (ii) *$p(\boldsymbol{N})$, the marginal distribution, is uniform*
- (iii) *$p(\tilde{\boldsymbol{N}}|\boldsymbol{N})$, the conditional distribution, is a rotationally asymmetric generalized normal distribution [51], i.e. a Laplace distribution.*
- (iv) *the observations are generated by a smooth, bijective (i.e., invertible) mapping*

*(v) the contrastive objective uses the same metric as $p(\tilde{\boldsymbol{N}}|\boldsymbol{N})$, i.e. $L_1$ for Laplace (cf. [63, Def. 1]).*

Table 3: Hyperparameters for our experiments (§ 4)

| PARAMETER | VALUES |
|---|---|
| $\widehat{\boldsymbol{f}}^{-1}$ | 6-LAYER MLP |
| ACTIVATION | LEAKY RELU |
| BATCH SIZE | 6144 |
| LEARNING RATE | 1e−4 |
| $\mathbb{R}^d$ | $[0;1]^d$ |
| $C_p$ | 1 |
| $m_p$ | 0 |
| $C_{param}$ | 0.05 |
| $m_{param}$ | 1 |
| $p$ | 1 |
| $\tau$ | 1 |
| $\alpha$ | 0.5 |

$N$ noise (independent) variable component
$X$ observation component
$\boldsymbol{N}$ noise (independent) variable vector
$\boldsymbol{Pa}$ parent set of $\boldsymbol{X}$
$\boldsymbol{X}$ observation vector
$\mathcal{A}$ adjacency matrix of a SEMs
$\mathcal{C}$ connectivity matrix of a SEMs
$\boldsymbol{f}$ structural assignment in SEMs
$\mathcal{I}$ index set
$\pi$ causal ordering
$f$ a component of $\boldsymbol{f}$

# C NOTATION

**ACRONYMS**
**ANM** Additive Noise Model

**CD** Causal Discovery
**CL** Contrastive Learning

**DAG** Directed Acyclic Graph
**DGP** Data Generating Process

**ICA** Independent Component Analysis
**ICM** Independent Causal Mechanisms

**LiNGAM** Linear Non-Gaussian Acyclic Model

**MCC** Mean Correlation Coefficient
**MLP** MultiLayer Perceptron

**NLICA** NonLinear Independent Component Analysis

**SEM** Structural Equation Model
**SHD** Structural Hamming Distance

**NOMENCLATURE**
$\alpha$ scalar field
$\mathbf{D}$ diagonal matrix
$\mathbf{I}_d$ $d$-dimensional identity matrix
$\mathbf{J}$ Jacobi matrix
$\mathcal{L}_{CL}$ contrastive loss function
$\mathcal{L}_\pi$ regularizer for learning $\pi$
$\mathbf{S}$ Sinkhorn network
$\mathcal{L}$ loss function
$\sim_{DAG}$ structural equivalence
$d$ problem dimensionality

**Causality**