# OpenReview forum: "Multivariable Causal Discovery with General Nonlinear Relationships"
_auai.org/UAI/2022/Workshop/CRL — CRL@UAI 2022 Oral_

### Official Review · Reviewer_YE7b · 2022-06-27
**Good idea + implementation; accept**

**Rating:** 7
**Confidence:** 5

**Review:**

This submission focuses on causal discovery in the presence of general, non-linear relationships. This is a challenging problem that has only recently been approached, and in fact many recent methods (e.g., Monti et al 2019) focus primarily on the bivariate case whereas the primary focus of this submission is over the more general, $d$ dimensional case (with $d > 2$).

As the authors note, the approach of Monti et al (2019) can in principle be scaled to arbitrary $d$, however, since their approach is based off first performing nonlinear ICA and then a series of independence tests, it cannot easily be applied to the multivariable case (for one, the large number of tests means that all statistical power will be lost if any multiple comparison strategy is employed).

Instead the authors focus on the Jacobian of the mapping from observations to latent sources. The insight of the paper is nice, noting that the Jacobian will correctly encode the sparse support of the associated DAG. The challenge remains that the ordering of variables can be arbitrarily permuted, as in the case in LiNGAM which the authors take as inspiration to propose a training objective to be minimized.

Overall the paper is well written. The experimental section is small but the results are clearly communicated.

Minor questions:
1. why do sinkhorn networks need to be used ? Would it not be possible to simply learn the f^{-1} (e.g., using nonlinear ICA or similar) and then study the properties of the Jacobian matrix ? (wouldnt this be more in line with LiNGAM, which first estimates ICA unmixing matrix and then permutes it?)

---

### Official Review · Reviewer_hjF4 · 2022-06-27
**Well-motivated, well-written paper solving an important problem with an interesting method, discussion and rigours proofs.**

**Rating:** 9
**Confidence:** 4

**Review:**

The work focuses on the multivariable causal discovery problem with general nonlinear functions in SEMs.

Firstly, it clearly introduced the challenges of causal discovery and its relationship with (NL-)ICA.

Secondly, to achieve the identifiability with general nonlinear functions in SEMs, it assumes the strong identifiability [24, Def.1]. Even though the assumption can not always hold and it would be even better with analyzing the cases where the strong identifiability doesn’t hold, the discussion and the method in work can be still interesting enough for the causal discovery community.

Moreover, given the results of [61], the paper further extracts the Jacobian matrix and proposes a LiNGAM-style causal discovery method for the multivariate case, which solves the scalability problem of [34] which is based on the bivariate case.

---

### Meta-Review · Program_Chairs · 2022-07-06

**Recommendation:** Accept (Oral)
**Confidence:** 4

**Metareview:**

Both reviewers were very positive about the paper, agreeing that it is well-written and interesting. Hence, I recommend this work to be accepted and highlighted (as a talk) at the workshop.

---

### Decision · Program_Chairs · 2022-07-06

Accept (Oral)